# Graph Bernoulli Pooling

## Abstract

Graph pooling is crucial for enlarging receptive field and reducing computational cost in deep graph representation learning. In this work, we propose a simple but effective non-deterministic graph pooling method, called graph Bernoulli pooling (BernPool), to facilitate graph feature learning. In contrast to most graph pooling methods with deterministic modes, we design a probabilistic Bernoulli sampling to reach an expected sampling rate through deducing a variational bound as the constraint. To further mine more useful info, a learnable reference set is introduced to encode nodes into a latent expressive probability space. Hereby the resultant Bernoulli sampling would endeavor to capture salient substructures of the graph while possessing much diversity on sampled nodes due to its non-deterministic manner. Considering the complementarity of node dropping and node clustering, further, we propose a hybrid graph pooling paradigm to combine a compact subgraph (via dropping) and a coarsening graph (via clustering), in order to retain both representative substructures and input graph info. Extensive experiments on multiple public graph classification datasets demonstrate that our BernPool is superior to various graph pooling methods, and achieves state-of-the-art performance. The code is publicly available in an anonymous format at https:/github/BernPool.

## 1  Introduction

Graph Neural Networks (GNNs) [13, 36] have been widely used to learn expressive representation from ubiquitous graph-structured data such as social networks [31], chemical molecules [15] and biological networks [24]. To improve representation ability, multiple GNN variants, e.g., graph convolutional networks (GCNs) [16] and graph attention networks (GATs) [25], have been developed to facilitate various graph-related tasks including node classification[16], link prediction[19, 35], and graph classification[36]. Specifically, for graph-related learning tasks, graph pooling has become an essential component in various GNN architectures. Aiming to learn compact representation for graphs, graph pooling facilitates graph topology modeling by enlarging receptive fields as well as scaling down the graph size which effectively reduces computational costs.

The existing graph pooling techniques generally fall into two main categories, i.e., the global graph pooling [5, 27, 14, 33, 26, 36] and hierarchical graph pooling. The former directly compresses a set of nodes into a compact graph-level representation. This operation results in a flat feature as a whole graph embedding. In contrast, the hierarchical pooling coarsens graphs gradually and outputs the corresponding pooled graphs of smaller sizes. For this purpose, two different types of coarsening, named node dropping [17, 10, 20, 37, 18, 11] and node clustering [33, 1, 34], are often employed. The node dropping picks up a subset of nodes to construct the coarsened graph, while the node clustering learns an assignment matrix to aggregate those nodes in the original graph into new clusters. In this work, our proposed BernPool falls in the category of the latter.

Even though considerable progress has been made, most pooling methods select a part of nodes or cluster nodes in a deterministic manner according to the importance scores of nodes, which degrades the sampling diversity. For this issue, some previous works [21, 7] propose stochastic node dropping independent from the data, i.e., randomly dropping. However, they do not consider the intrinsic

Submitted to 37th Conference on Neural Information Processing Systems (NeurIPS 2023). Do not distribute.

structural characteristics of data while enriching the sampling diversity. Thus the current bottleneck is how to adaptively extract expressive substructures and meanwhile keeping rich sampling diversity in graph pooling, with the precondition of high-efficient and effective graph representation learning.

To address the above problem, in this work, we propose a graph Bernoulli pooling method called BernPool to facilitate graph representation learning. Different from the existing graph pooling methods, we design a probabilistic Bernoulli sampling by estimating the sampling probabilities of graph nodes. To restrict the sampling process, we formulate a variational Bernoulli learning constraint by deriving an upper bound between an expected distribution and a learned distribution. To better capture expressive info, a learnable reference set is further introduced to encode nodes into a latent expressive probability space. Thus the advantage of the resultant Bernoulli sampling is two-fold: i) capture representative substructures of graph; and ii) preserve certain diversity (like random dropping) due to its non-deterministic manner. Considering the complementary characteristics between node dropping and node clustering, we propose a hybrid graph pooling paradigm to fuse a compact subgraph after node dropping and a coarsening graph after node clustering. The node clustering in our framework is also Bernoulli-induced without high-computation cost because it adopts the sampled nodes as clustering centers. The hybrid graph pooling can jointly learn representative substructures and preserve the input graph topology. We conduct extensive experiments on 8 public graph classification datasets to test our BernPool, and the experimental results validate that our BernPool achieves better performance than those existing pooling methods and keep high efficiency on par with those node-dropping methods.

The contributions of this work are summarized as: i) propose a probabilistic Bernoulli sampling method to not only learn effective sampling but also preserve high efficiency; ii) propose a hybrid graph pooling way to retain both those sampled substructures and the remaining info; iii) verify the effectiveness and high-efficiency of our BernPool, and report the state-of-the-art performance.

## 2   Related Work

In this section, we first review the previous methods of Graph Neural Networks (GNNs), then introduce the related Hierarchical pooling methods.

**Graph Neural Network.** GNNs were introduced as a form of recurrent neural network by Gori et.al. [12] and Scarselli et al. [23]. Subsequently, Duvenaud et al.[6] introduced a convolution-like propagation rule on graphs to extract node representations for graph-level classification. To enhance the graph representation ability, several convolution operations were proposed (e.g. Graph Convolution Network(GCN[16]), Graph Attention Network(GAT[25]), GraphSAGE[13], GIN[30]) to extract expressive node representations by aggregating neighbor node features and have achieved promising performance in various graph-related tasks in recent years. In particular, GCN[16] utilizes a first-order approximation of spectral convolution via Chebyshev polynomial iteration to improve efficiency. However, it suffers from the issue of fixed and equal weighting for neighbor nodes during aggregation, which may not be optimal for all nodes and can lead to information loss. To address this problem, GAT[25] introduced an attention mechanism to assign different weights to neighbor nodes during message passing. Furthermore, GrapSAGE[13] learned node embeddings by aggregating feature information from the neighborhood in an inductive manner. Despite the considerable progress made by GNNs, they are limited in their ability to generate hierarchical graph representations due to the lack of pooling operations.

**Hierarchical Graph Pooling.** Graph pooling is a critical operation to obtain robust representations and scale down the size of graphs, which can be classified into two categories: global pooling and hierarchical pooling. The former [5, 27, 14, 33, 26, 36] aggregates node-level features to generate a graph-level representation. For instance, SortPool[36] ranks and groups the nodes into clusters according to their features, then aggregates the resulting clusters to generate the graph-level representation. However, the global pooling suffers from the issue of discarding structure information in generating graph-level representation. On the other hand, the hierarchical pooling methods could progressively compress the graph into a smaller one and capture the hierarchical structure. They can be further divided into node clustering pooling methods [33, 1, 34], node drop pooling methods [17, 10, 20, 37, 18, 11] and other pooling methods[28, 3]. Among them, the node drop pooling methods deleted the unimportant nodes based on certain criteria. For instance, SAGPool [17] computed the node attention scores using graph convolution to preserve the most

important nodes. But the node drop pooling methods may not preserve the original structure well during the graph compression process. To alleviate this problem, Pang et.al [20] applied contrastive learning to maximize the mutual information between the input graph and the pooled graphs to preserve the graph-level dependencies in the pooling layers. Gao et.al [11] proposed a criterion to assess each node's information among its neighbors to retain informative features. Our BernPool introduces probabilistic deduced Bernoulli sampling based on reference set to progressively compress the original graph by preserving important nodes, rather than selecting nodes in a deterministic way. This probabilistic manner can lead to more diverse sampling situations and capture the data of intrinsic characteristics, promoting graph discriminative representation learning. Furthermore, we propose a hybrid graph pooling module to alleviate the node drop pooling method's issue of cannot preserve structure well.

## 3   Preliminaries

**Notations**   For an arbitrary graph $\mathcal{G} = (\mathcal{V}, \mathcal{E}, \mathbf{X})$ with $n = |\mathcal{V}|$ nodes and $|\mathcal{E}|$ edges. $\mathbf{X} \in \mathbb{R}^{n \times d'}$ represents the node feature matrix, where $d'$ is the dimension of node attributes, and $\mathbf{A} \in \mathbb{R}^{n \times n}$ denotes the adjacency matrix describing its edge connection information. The graph $\mathcal{G}$ has a one-hot vector $\mathbf{y}_i$ w.r.t its label. A pooled graph of the original graph $\mathcal{G}$ is denoted by $\widetilde{\mathcal{G}} = (\widetilde{\mathcal{V}}, \widetilde{\mathcal{E}}, \widetilde{\mathbf{X}})$ with adjacency matrix as $\widetilde{\mathbf{A}} \in \mathbb{R}^{\widetilde{n} \times \widetilde{n}}$, where $\widetilde{n}$ denotes the number of pooled nodes.

**Graph Convolution**   In this work, we employ the classic graph convolution network (GCN) as the backbone to extract features, where the $l$-th convolutional layer is formulated as:

$$\mathbf{H}^{(l+1)} = \sigma(\hat{\mathbf{D}}^{-\frac{1}{2}} \hat{\mathbf{A}} \hat{\mathbf{D}}^{\frac{1}{2}} \mathbf{H}^{(l)} \mathbf{W}^{(l)}), \tag{1}$$

where $\sigma(\cdot)$ is a non-linear activation function, $\mathbf{H}^{(l)}$ is the hidden-layer feature, $\hat{\mathbf{A}}$ is the added self-loop adjacent matrix, $\hat{\mathbf{D}}$ denotes the degree matrix of $\hat{\mathbf{A}}$, and $\mathbf{W}^{(l)}$ represents a learnable weight matrix at the $l$-th layer. The initial node features are used at the first convolution, i.e., $\mathbf{H}^{(0)} = \mathbf{X}$.

## 4   The Proposed BernPool

### 4.1   Overview

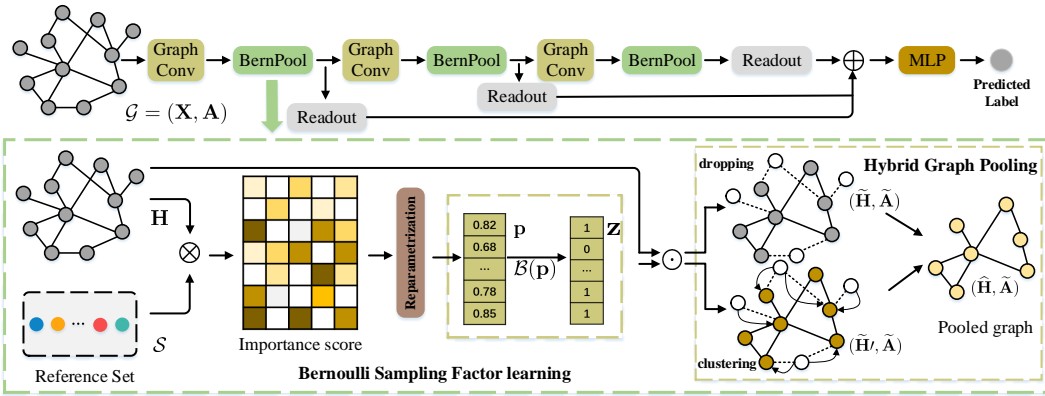

Figure 1: The architecture of the proposed BernPool framework. Please see Overview in Section 4.1.

The whole framework is illustrated in Fig. 1, where the convolution and our proposed BernPool are stacked alternately. The proposed BernPool could be seamlessly engaged with any type of graph convolution to facilitate graph representation. In BernPool, there contains two main modules: graph Bernoulli sampling (GBS) and hybrid graph pooling (HGP). To boost graph Bernoulli sampling, we specifically design a reference set $\mathcal{S}$ to encode the importance of each graph node, and the reference set is configured as the optimizable parameters. The reference set is further conditioned on the orthogonal space so as to reduce redundancy. After transforming via the reference set, we can

estimate the sampling probabilities $\mathbf{p}$ of graph nodes. In particular, the probabilities of nodes are totally restricted with an expected/predefined distribution, which is formulated to maximize the upper bound of KL-divergence (please see Section 4.2). Just due to the restriction, we can probabilistically sample a specified proportion of graph nodes. The detail of GBS can be found in Section 4.3. In the stage of hybrid graph pooling, on the one hand, we prune those unsampled nodes and the associated edges to generate a compact subgraph; on the other hand, to preserve graph topological structure, we perform neighbor nodes clustering to form a coarsening graph. Both the compact subgraph and coarsening graph are fused to form the final pooled graph. The detail of HGP can be found in Section 4.4. The BernPool attempts to learn the reference set and a few linear transformations. The whole framework can be optimized in an end-to-end mode through back-propagation.

## 4.2 Bernoulli Sampling Optimization Objective

To sample a certain proportion of graph nodes in a probabilistic manner, we derive a KL-divergence constraint, which makes node probabilities tend to be a predefined distribution. To this end, we again dissect mutual information between the learned subgraph embeddings and their corresponding labels. Formally, we aim to maximize the mutual information function:

$$\zeta_{MI} = MI(\mathbf{y}, f_{\psi,\phi}(\mathcal{G}, \mathcal{S})), \tag{2}$$

where $\phi$ denotes the parameters of the BernPool, $\psi$ is the parameters of other modules (e.g., convolution, classifier), and $f_{\psi,\phi}(\cdot)$ represents the graph embedding process.

Suppose the sampling factor $\mathbf{z}$ in graph pooling, we resort to the relationship between mutual information and expectation, and rewrite Eqn. (2) as:

$$\mathbb{E}_{\mathbf{y}|\mathcal{G},\mathcal{S}}[\log \int p_{\psi}(\mathbf{y}|\mathcal{G}, \mathcal{S}, \mathbf{z}) p_{\phi}(\mathbf{z}|\mathcal{G}, \mathcal{S}) d\mathbf{z}]$$
$$= \mathbb{E}_{\mathbf{y}|\mathcal{G},\mathcal{S}}[\log \int q_{\phi}(\mathbf{z}|\mathcal{G}, \mathcal{S}) p_{\psi}(\mathbf{y}|\mathcal{G}, \mathcal{S}, \mathbf{z}) \frac{p_{\phi}(\mathbf{z}|\mathcal{G}, \mathcal{S})}{q_{\phi}(\mathbf{z}|\mathcal{G}, \mathcal{S})} d\mathbf{z}], \tag{3}$$

where $p_{\psi}(\mathbf{y}|\mathcal{G}, \mathcal{S}, \mathbf{z})$ is the conditional probability of label $\mathbf{y}$, and $p_{\phi}(\mathbf{z}|\mathcal{G}, \mathcal{S})$ denotes the conditional probability of the factor $\mathbf{z}$ that is usually intractable. After a series of derivation from Eqn. (3), we can deduce a bound with KL-divergence between expected Bernoulli distribution $q_{\phi}$ and learned distribution $p_{\phi}$:

$$\mathbb{E}_{\mathbf{y}|\mathcal{G},\mathcal{S}}[\log \int p_{\psi}(\mathbf{y}|\mathcal{G}, \mathcal{S}, \mathbf{z}) p_{\phi}(\mathbf{z}|\mathcal{G}, \mathcal{S}) d\mathbf{z}]$$
$$\geq \mathbb{E}_{\mathbf{y}|\mathcal{G},\mathcal{S}}[\log p_{\psi}(\mathbf{y}|\mathcal{G}, \mathcal{S}, \mathbf{z})] - D_{KL}(q_{\phi}(\mathbf{z}|\mathcal{G}, \mathcal{S})||p_{\phi}(\mathbf{z}|\mathcal{G}, \mathcal{S})) \tag{4}$$
$$= -\zeta_{CE} - D_{KL}(q_{\phi}(\mathbf{z}|\mathcal{G}, \mathcal{S})||p_{\phi}(\mathbf{z}|\mathcal{G}, \mathcal{S})), \tag{5}$$

where $p_{\psi}(\mathbf{y}|\mathcal{G}, \mathcal{S})$ represents the predicted probability of label based on the input graph and reference set, $\zeta_{CE}$ is the cross entropy loss function. **Please see the detailed derivation in the supplementary file.** As the expected Bernoulli distribution $q_{\phi}(\mathbf{z}|\mathcal{G}, \mathcal{S})$ is independent from the input graph and reference set, $q_{\phi}(\mathbf{z}|\mathcal{G}, \mathcal{S})$ can be denoted as $q(\mathbf{z})$. After adding the soft-orthogonal constraint on the reference set, therefore, the final optimization objective can be converted to minimize:

$$\zeta = \zeta_{CE} + D_{KL}(q(\mathbf{z})||p_{\phi}(\mathbf{z}|\mathcal{G}, \mathcal{S})) + \beta||\mathbf{S}\mathbf{S}^{\mathsf{T}} - c\mathbf{I}||_{F}, \tag{6}$$

where the second term forces the sampling factor to follow an expected distribution $q$, the matrix $\mathbf{S}$ stacks the vectors of reference set $\mathcal{S}$ in the third term, $\mathbf{I}$ denotes the identity matrix, $\beta$ is a trade-off hyper-parameter, and $c$ is a learnable scalar. Thus, the learning of the sampling factor could be integrated into the objective function as a joint training process. In addition, we can easily extend the above single-layer BernPool into multi-layer networks by deploying independent sampling factors in sequential graph pooling.

## 4.3 Bernoulli Sampling Factor Learning

To extract an expressive sub-graph $\widetilde{\mathcal{G}}$ from the original or former-layer graph $\mathcal{G}$, we estimate a probabilistic factor $\mathbf{z} = (z_1, \cdots, z_n)^{\mathsf{T}} \in \{0, 1\}^n$ that conforms to Bernoulli distribution, instead of a deterministic way such as top-k. In contrast to the deterministic way, our BernPool possesses

a more diverse sampling in mining substructures and graph topological variation of input data. However, as mentioned above in Eqn. (3), it's rather non-trivial to infer $\mathbf{z}$ through Bayes rule: $p(\mathbf{z}|\mathcal{G}, \mathcal{S}) = p(\mathbf{z})p(\mathcal{G}, \mathcal{S}|\mathbf{z})/p(\mathcal{G}, \mathcal{S})$. A reason is that the prior $p(\mathcal{G}, \mathcal{S}|\mathbf{z})$ is intractable. We resort to the variational inference to approximate the intractable true posterior $p(\mathbf{z}|\mathcal{G}, \mathcal{S})$ with $q(\mathbf{z})$ by constraining the KL-divergence $D_{KL}(q_\phi(\mathbf{z})||p_\phi(\mathbf{z}|\mathcal{G}, \mathcal{S}))$.

To make the gradient computable, we employ the reparameterization trick to derive $p(\mathbf{z}|\mathcal{G}, \mathcal{S}) = \mathcal{B}(\mathbf{p})$, where $\mathcal{B}(\cdot)$ denotes the Bernoulli distribution based on the probability vector. Concretely, we take the graph convolution feature $\mathbf{H}$ and the matrix $\mathbf{S}$ of reference set as input, to learn the sampling probability $\mathbf{p}$. A simple formulation with a fully-connected operation is given as follows:

$$\mathbf{p} = \text{sigmoid}(\text{MLP}(\cos(\mathbf{H}, \mathbf{S}))), \tag{7}$$

where $\cos(\mathbf{H}, \mathbf{S})$ computes the pairwise similarities (across all graph nodes and all reference points) through the cosine measurement, and $\text{sigmoid}(\cdot)$ is the sigmoid function.

### 4.4 Bernoulli Hybrid Graph Pooling

Based on the above inferred sampling factor $\mathbf{z}$, we propose a hybrid graph pooling to learn expressive substructures while endeavoring to reserve graph information. The hybrid graph pooling contains two components: Bernoulli node dropping and Bernoulli node clustering. The former prunes those unsampled nodes and associated edges to generate a compact graph; the latter clusters neighbor nodes to form a coarsening graph.

**Bernoulli Node Dropping.** Let $\text{idx} \in \mathbb{R}^{\widetilde{n}}$ denotes the indices of the preserved nodes according to the sampling factor $\mathbf{z}$, thus a projection matrix $\mathbf{P} \in \{0, 1\}^{\widetilde{n} \times n}$ can be defined formally:

$$\mathbf{P} = \text{diag}(\mathbf{z})[\text{idx}, :], \tag{8}$$

where $\text{diag}(\cdot)$ is the vector diagonalization operation, and $\mathbf{X}[\text{idx}, :]$ extracts those rows of $\mathbf{X}$ w.r.t the indices $\text{idx}$. Accordingly, the compact subgraph $\widetilde{\mathcal{G}} = (\widetilde{\mathbf{H}}, \widetilde{\mathbf{A}})$ can then be computed by:

$$\widetilde{\mathbf{H}} = \mathbf{PH}, \qquad \widetilde{\mathbf{A}} = \mathbf{PAP}^\intercal, \tag{9}$$

where $\widetilde{\mathbf{H}} \in \mathbb{R}^{\widetilde{n} \times d}$ is the feature matrix of the compact graph, and $\widetilde{\mathbf{A}} \in \mathbb{R}^{\widetilde{n} \times \widetilde{n}}$ is the subgraph adjacency matrix. Intuitively, the dropping directly removes those unselected nodes and connective edges from the input graph.

**Bernoulli Node Clustering.** After Bernoulli sampling, we can obtain those reserved nodes, which could be used as the clustering centers. Hereby, we only need to transmit those unselected nodes' messages to cluster centers, which would preserve more information of the whole input graph. Compared with previous graph clustering methods, Bernoulli clustering is more efficient because the learning of the assignment matrix is bypassed, while increasing the diversity of graph perception. To be specific, our assignment matrix $\mathbf{P}' \in \mathbb{R}^{\widetilde{n} \times n}$ is from the original adjacency matrix $\mathbf{A}$, and we can get the coarsening graph as:

$$\mathbf{P}' = (\text{diag}(\mathbf{z})[\text{idx}, :]) \times \mathbf{A}, \qquad \widetilde{\mathbf{H}}' = \mathbf{P}'\mathbf{H}, \tag{10}$$

where $\widetilde{\mathbf{H}}' \in \mathbb{R}^{\widetilde{n} \times d}$ denotes the diffusion features based on input graph, the coarsening graph has the same adjacency matrix $\widetilde{\mathbf{A}}_i$ as in Eqn. (9).

**Hybrid Graph.** Based on the aforementioned operations, we can obtain the compact subgraph $\widetilde{\mathcal{G}} = (\widetilde{\mathbf{H}}, \widetilde{\mathbf{A}})$ and the coarsening graph $\widetilde{\mathcal{G}}' = (\widetilde{\mathbf{H}}', \widetilde{\mathbf{A}})$, which reflect different aspects of information in the original graph. To fully exploit the extracted informative node representations and expressive substructures, the two subgraphs are fused to form the final pooled graph by the following formulation:

$$\widehat{\mathbf{H}} = \sigma((\widetilde{\mathbf{H}} + \widetilde{\mathbf{H}}')\mathbf{W}_h), \tag{11}$$

where $\sigma$ denotes a non-linear activation function and $\mathbf{W}_h \in \mathbb{R}^{d \times d}$ is a learnable weight, and $\widehat{\mathbf{H}} \in \mathbb{R}^{\widetilde{n} \times d}$ is the aggregated node embeddings of the pooled graph.

## 4.5 Readout Function

The proposed framework repeats the graph convolution and BernPool operations three times. To obtain a fixed-size graph-level representation, we apply the concatenation of max-pooling and mean-pooling in each subgraph following the previous works [37, 29, 22]. Finally, those graph-level representations can be summarized to form the final embeddings:

$$\mathbf{r} = \sum_{l=1,\cdots} \mathbf{r}^{(l)}, \quad \text{and} \quad \mathbf{r}^{(l)} = \sigma\left(\frac{1}{n_i^{(l)}}\sum_{i=1}^{n_i^{(l)}} \widehat{\mathbf{H}}_i^{(l)} \| \max_{i=1}^{n_i^{(l)}} \widehat{\mathbf{H}}_i^{(l)}\right), \tag{12}$$

where $\widehat{\mathbf{H}}_i^{(l)}$ denotes the $i$-th node feature at the $l$-th pooling, $\sigma$ is the same non-linear activation function, and $\|$ denotes the feature concatenation operation. The resulting embeddings would finally be fed into a multi-layer perceptron to predict graph labels.

## 4.6 Computational Complexity

The computational complexity of one-layer BernPool can be expressed as $O(N \times K \times d + N' \times N \times d + N' \times d \times d)$, where $N$ denotes the number of nodes, $K$ is the number of reference points, d is the dimensionality of nodes features, $N'$ represents the number of preserved nodes. Specifically, the complexity of probability score computation is $O(N \times K \times d)$, and the complexity of the Bernoulli hybrid graph pooling module is $O(N' \times N \times d + N' \times d \times d)$.

# 5 Experiments

## 5.1 Experimental Setup

**Datasets**. To comprehensively evaluate our proposed model, we conduct extensive experiments on eight widely used datasets in the graph classification task, including three social network datasets (IMDB-BINARY, IMDB-MULTI [31] and COLLAB[32]) and five Bioinformatics datasets (PROTEINS[8], DD[4], NCI1[24], Mutagenicity[15] and ENZYMES[2]). The detailed information and statistics of these datasets are summarized in Table. 1.

**Baselines**. We compare our proposed method with several state-of-the-art graph pooling methods, including three backbones (GCN, GAT, GraphSAGE), eight node drop graph pooling methods (TopkPool[10], SAGPool[17], ASAP[22], VIPool[18], iPool[11], CGIPool[20], SEP-G [29] and MVPool [37]), three clustering pooling methods (Diffpool[33], MincutPool[1], and StructPool[34]), three global pooling methods (Set2Set[26], SortPool[36], DropGIN[21]) and one other pooling method (EdgePool[3]).

**Implementation Details**. We employ the 10-fold cross-validation protocol following the settings of [29, 22] and report the average classification accuracies and standard deviation. For all used datasets, we set the expected pooling ratio as 0.8, the node embedding dimension $d$ as 128, the number of reference points as 32, and the hyper-parameter $\beta$ in Eqn. 6 as 5. We adopt the Adam optimizer to train our model with 1000 epochs, where the learning rate and weight decay are set as 1e-3 and 1e-4, respectively. Our proposed BernPool is implemented with PyTorch and Pytorch Geometric [9].

## 5.2 Comparison with the state-of-the-art Methods

The graph classification results of BernPool and other state-of-the-art methods are presented in Table 1. In general, most hierarchical pooling approaches including our proposed BernPool can perform better than those global pooling ones in the graph classification task. This may be because global pooling methods ignore the hierarchical graph structures in generating graph-level representation. In particular, our BernPool achieves state-of-the-art performance on all datasets, which demonstrates the robustness of our framework against graph structure data variation. In contrast, previous methods cannot perform well on all eight datasets, while the second highest performances on different datasets are obtained by five different methods. Compared with those methods, Our BernPool outperforms respectively by 1.09%, 3.16%, 13.6%, 2.7%, and 1.86% on the PROTEINS, Mutagenicity, ENZYMES, IMDB-BINARY and COLLAB datasets.

Table 1: **The statistics of eight datasets and graph classification accuracies comparison of different methods.** We have highlighted the best results in black and marked the second-highest accuracies with the symbol †.

| | Bioinformatics | | | | | Social Network | | |
|---|---|---|---|---|---|---|---|---|
| | PROTEINS | DD | NCI1 | Mutagenicity | ENZYMES | IMDB-B | IMDB-M | COLLAB |
| #Graphs(Classes) | 1113 (2) | 1178 (2) | 4110 (2) | 4337 (2) | 600 (6) | 1000 (2) | 1500 (3) | 5000 (3) |
| Avg # Nodes | 39.1 | 284.3 | 29.8 | 30.3 | 32.6 | 19.8 | 13.0 | 74.5 |
| Avg # Edges | 72.8 | 715.7 | 32.3 | 30.8 | 62.1 | 96.5 | 65.9 | 2457.8 |
| GCN[16] | 74.84±2.82 | 78.12±4.33 | 76.3±1.8 | 79.8±1.6 | 50.00±5.87 | 72.67±6.42 | 50.40±3.02 | 71.92±3.24 |
| GAT[25] | 74.07±4.53 | 75.56±3.72 | 74.9±1.7 | 78.8±1.2 | 51.00±5.23 | 74.07±4.53 | 49.67±4.30 | 75.80±1.60 |
| GraphSAGE[13] | 73.75±2.97 | 77.27±4.06 | 74.7±1.3 | 78.9±2.1 | 53.33±3.42 | 72.17±5.29 | 48.53±5.43 | 79.70±1.70 |
| Set2Set[26] | 73.27±0.85 | 71.94±0.56 | 68.55±1.92 | 71.35±2.1 | - | 72.90±0.75 | 50.19±0.39 | 79.55±0.39 |
| SortPool[36] | 73.27±0.85 | 75.58±0.72 | 73.82±1.96 | 70.66±1.51 | 49.67±4.27 | 72.12±1.12 | 48.18±0.83 | 77.87±0.47 |
| DropGIN[21] | 76.3±6.1 | - | - | - | - | 75.7±4.2 | 51.4±2.8 | - |
| EdgePool[3] | 72.50±3.2 | 75.85±0.58 | - | - | - | 72.46±0.74 | 50.79±0.59 | 67.10±2.7 |
| DiffPool[33] | 73.03±1.00 | 77.56±0.41 | 62.32±1.90 | 77.60±2.70 | 61.83±5.3 | 73.14±0.70 | 51.31±0.72 | 78.68±0.43 |
| StructPool[34] | 80.36 | 84.19 | - | - | 63.83 | 74.70 | 52.47 | 74.22 |
| MinCutPool[1] | 76.5±2.6 | 80.8±2.3 | 74.25±0.86 | 79.9±2.1 | - | 72.65±0.75 | 51.04±0.70 | 83.4±1.7† |
| TopKPool[10] | 70.48±1.01 | 73.63±0.55 | 67.02±2.25 | 79.14±0.76 | 50.33±6.3 | 71.58±0.95 | 48.59±0.72 | 77.58±0.85 |
| SAGPool[17] | 71.86±0.97 | 76.45±0.97 | 67.45±1.11 | 72.40±2.40 | 52.67±5.8 | 72.55±1.28 | 50.23±0.44 | 78.03±0.31 |
| ASAP[22] | 74.19±0.79 | 76.87±0.70 | 71.48±0.42 | 80.12±0.88 | - | 72.81±0.50 | 50.78±0.75 | 78.64±0.50 |
| VIPool[18] | 79.91±4.1 | 82.68±4.1† | - | 80.19±1.02 | 57.50±6.1 | 78.60±2.3† | 55.20±2.5† | 78.82±1.4 |
| iPool[11] | 76.46±3.22 | 78.76±3.45 | 80.46±1.66† | - | 56.00±7.72 | 72.90±3.08 | 50.73±3.68 | 76.86±1.67 |
| SEP-G[29] | 76.42±0.39 | 77.98±0.57 | 78.35±0.33 | - | - | 74.12±0.56 | 51.53±0.65 | 81.28±0.15 |
| CGIPool[20] | 74.10±2.31 | - | 78.62±1.04 | 80.65±0.79† | - | 72.40±0.87 | 51.45±0.65 | 80.30±0.69 |
| MVPool[37] | 82.2±1.2† | 78.4±1.5 | 77.5±1.3 | 80.2±0.8 | 62.4±2.5† | - | 48.6±1.0 | - |
| BernPool(Ours) | 83.29 ± 3.69 | 83.27 ± 2.95 | 81.44 ± 1.09 | 83.81 ± 1.43 | 76.00 ± 3.78 | 81.30 ± 3.50 | 55.93 ± 3.8 | 85.26 ± 1.35 |

Moreover, the proposed BernPool significantly surpasses GNNs (GCN, GAT, GraphSAGE) without adding the pooling operation, which verifies our BernPool can effectively learn representative substructures and preserve graph topological information. SEP-G [29] gains better performance compared with other node drop pooling methods, which utilizes the structural entropy to assess the importance of each graph node. However, our BernPool exhibits an average 5% relative improvement over SEP-G. This can be attributed to our method of learning sampling factors in a probabilistic manner, which possesses more diversity in mining expressive sub-structures compared with the deterministic way. Compared to DiffPool [33], our BernPool outperforms it on all used datasets, verifying the effectiveness of our proposed hybrid graph pooling module, which jointly leverages the advantages of both node drop and clustering methods. Particularly, DropGIN [21] randomly drops nodes in the training process, which is also a probabilistic manner. Our BernPool outperforms it by 6.99%, 5.6%, and 4.53% on the PROTEINS, IMDB-BINARY, and IMDB-MULTI datasets. This is because our BernPool considers the characteristics of data and can adaptively handle the topology variations.

# 6 Ablation Study

**Effectiveness of the proposed BernPool using GCN variants.** To evaluate the performance of our method by employing different convolution layers, we integrate three widely used ones (i.e., GCN[16], GAT[25] and GraphSAGE[13]) into our BernPool framework. The results on eight datasets are shown in Table 2. It can be observed that all three variants (BernPool-GCN, BernPool-GAT, BernPool-GraphSAGE) outperform their corresponding backbones, significantly improving the graph classification performances. This observation validates the effectiveness of the proposed BernPool.

**Effectiveness of Bernoulli hybrid graph pooling.** To verify the effectiveness of our proposed Bernoulli hybrid graph pooling, we further conduct experiments on eight datasets. As the hybrid graph pooling consists of node dropping and clustering. We separately remove one of the channels and keep the other parts the same. We name the BernPool without dropping and clustering as "BernPool w/o Dropping" and "BernPool w/o Clustering", respectively. The detailed results are reported in Table 3.

Table 2: **Graph classification accuracies of BernPool using different backbones.** The default backbone is GCN.

| Variants | Bioinformatics | | | | | Social Network | | |
|---|---|---|---|---|---|---|---|---|
| | PROTEINS | DD | NCI1 | Mutagenicity | ENZYMES | IMDB-B | IMDB-M | COLLAB |
| #Graphs(Classes) | 1113 (2) | 1178 (2) | 4110 (2) | 4337 (2) | 600 (6) | 1000 (2) | 1500 (3) | 5000 (3) |
| GCN | 74.84±2.82 | 78.12±4.33 | 76.3±1.8 | 79.8±1.6 | 50.00±5.87 | 72.67±6.42 | 50.40±3.02 | 71.92±3.24 |
| BernPool-GCN | 83.29±3.69 ↑ | 83.27±2.95 ↑ | 81.44±1.09 ↑ | 83.81±1.43 ↑ | 76.00±3.78 ↑ | 81.30±3.5 ↑ | 55.93±3.8 ↑ | 85.26±1.35 ↑ |
| GAT | 74.07±4.53 | 75.56±3.72 | 74.9±1.7 | 78.8±1.2 | 51.00±5.23 | 74.07±4.53 | 49.67±4.30 | 75.80±1.60 |
| BernPool-GAT | 81.49±3.81 ↑ | 83.44±3.57 ↑ | 81.29±1.77 ↑ | 84.34±1.58 ↑ | 69.50±5.45 ↑ | 81.20±3.39 ↑ | 55.00±4.47 ↑ | 83.86±1.57 ↑ |
| GraphSAGE | 73.75±2.97 | 77.27±4.06 | 74.7±1.3 | 78.9±2.1 | 53.33±3.42 | 72.17±5.29 | 48.53±5.43 | 79.70±1.70 |
| BernPool-GraphSAGE | 83.20±4.21 ↑ | 82.25±3.49 ↑ | 82.34±1.61 ↑ | 84.67±1.26 ↑ | 75.67±3.78 ↑ | 81.60±3.60 ↑ | 55.47±3.87 ↑ | 84.56±1.02 ↑ |

Table 3: **Performance Comparison between BernPool and its variants.**

| Variants | Bioinformatics | | | | | Social Network | | |
|---|---|---|---|---|---|---|---|---|
| | PROTEINS | DD | NCI1 | Mutagenicity | ENZYMES | IMDB-B | IMDB-M | COLLAB |
| #Graphs(Classes) | 1113 (2) | 1178 (2) | 4110 (2) | 4337 (2) | 600 (6) | 1000 (2) | 1500 (3) | 5000 (3) |
| BernPool w/o Clustering | 81.94±4.27 | 81.66±2.82 | 78.22±7.69 | 82.57±1.44 | 73.00±3.67 | 80.60±3.13 | 55.80±4.11 | 84.52±1.14 |
| BernPool w/o Dropping | 82.40±4.02 | 82.08±3.98 | 76.20±10.24 | 82.75±1.44 | 72.50±4.25 | 81.30±3.23 | 55.60±4.16 | 85.06±1.18 |
| BernPool | 83.29±3.69 | 83.27 ± 2.95 | 81.44 ± 1.09 | 83.81 ± 1.43 | 76.00 ± 3.78 | 81.30 ± 3.5 | 55.93 ± 3.8 | 85.26 ± 1.35 |

Table 4: **Performance comparison between deterministic and probabilistic manner.**

| Variants | Bioinformatics | | | | | Social Network | | |
|---|---|---|---|---|---|---|---|---|
| | PROTEINS | DD | NCI1 | Mutagenicity | ENZYMES | IMDB-B | IMDB-M | COLLAB |
| #Graphs(Classes) | 1113 (2) | 1178 (2) | 4110 (2) | 4337 (2) | 600 (6) | 1000 (2) | 1500 (3) | 5000 (3) |
| BernPool-TopK | 81.77±3.53 | 82.09±3.25 | 81.09±1.77 | 83.26±1.08 | 68.17±3.72 | 80.20±3.74 | 55.47±3.51 | 84.50±1.40 |
| BernPool | 83.29±3.69 | 83.27 ± 2.95 | 81.44 ± 1.09 | 83.81 ± 1.43 | 76.00 ± 3.78 | 81.30 ± 3.5 | 55.93 ± 3.8 | 85.26 ± 1.35 |

Notably, our BernPool employs just one channel can achieve good results, which demonstrates the effectiveness of BernPool leveraging a probabilistic manner to infer sampling factors. We can observe that the "BernPool w/o Dropping" outperforms "BernPool w/o Clustering" by 0.46%, 0.42%, 0.70%, and 0.54% on PROTEINS, DD, IMDB-BINARY, and COLLAB datasets, respectively. Furthermore, jointly using both channels can outperform either "BernPool w/o Dropping" or "BernPool w/o Clustering", which verifies the effectiveness of our proposed hybrid graph pooling.

**Comparison between the probabilistic and deterministic manner.** To make clear the benefit of our proposed probabilistic sampling method, we conduct experiments by replacing Bernoulli sampling with Topk pooling which is in a deterministic manner. In the TopK experiments, we still employ reference set to assess the importance of nodes and the hybrid graph pooling module to jointly learn representative sub-structures and preserve graph topological information. The comparison between "BernPool-TopK" and "BernPool" is presented in Table 4. It can be observed that BernPool outperforms BernPool-TopK on all used eight datasets, especially 7.83% accuracy gains in the ENZYMES dataset, verifying the effectiveness of our designed Bernoulli-deduced sampling strategy.

**Benefit of the orthogonality for reference set.** To evaluate the benefit of orthogonality for the reference set, we conduct experiments that remove the orthogonal constraint from BernPool (referred to as "BernPool w/o orthogonal" in Fig. 2(a)). The results demonstrate that employing the orthogonal reference set can improve average accuracy by more than 0.6% on three datasets. This verifies the effectiveness of the orthogonal constraint for the reference set.

**Model parameter quantity comparison.** We compare the test accuracy and parameter quantity (only the pooling layer) of BernPool with other pooling methods in the PROTEINS dataset, where the hidden layer dimension is set to 128. BernPool achieves superior performance while using fewer parameters. Specifically, BernPool owns 97% fewer parameters than CGIPool and 76% fewer parameters than ASAP. In terms of accuracy, BernPool performs the best, achieving 9.19% higher than CGIPool and 9.10% higher accuracy than ASAP.

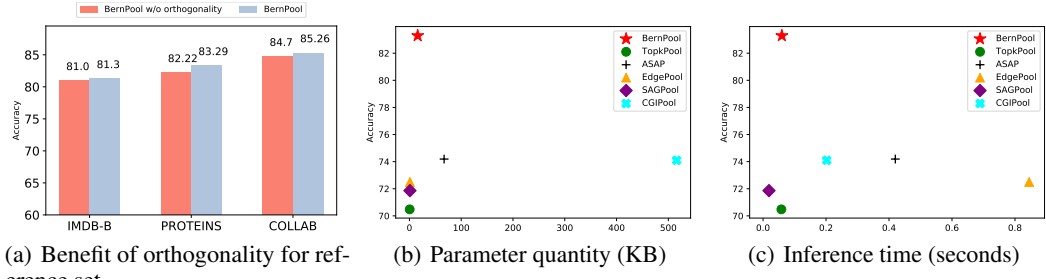

(a) Benefit of orthogonality for reference set

(b) Parameter quantity (KB)

(c) Inference time (seconds)

Figure 2: The benefit of orthogonality reference set and comparison of parameter quantity and inference time.

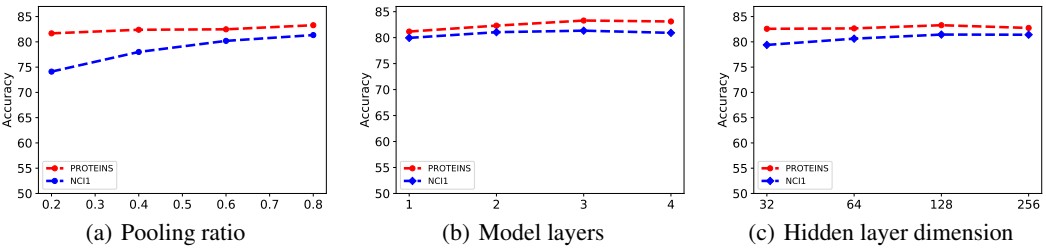

(a) Pooling ratio

(b) Model layers

(c) Hidden layer dimension

Figure 3: The performance of different hyper-parameters on PROTEINS and NCI1 datasets.

**Inference time comparison.** The computation complexity of our BernPool is described in Section 4.6. Moreover, the inference time also plays a crucial role in evaluating the efficiency of pooling methods. Thus we conduct experiments on the PROTEINS dataset containing about 20,000 nodes and 70,000 edges to compare the inference time (single-layer). As shown in Fig. 2(c), BernPool costs less inference time than EdgePool and ASAP while maintaining superior performance. Notably, SAGPool has a similar inference time as our method, but our method outperforms it in terms of classification accuracy. The comparison results verify the high efficiency of our BernPool.

**Sensitivity of Hyper-parameters.** To evaluate the sensitivity of hyper-parameters, including the pooling ratio, layer number, and hidden layer dimension, we additionally conduct experiments on PROTEINS and NCI1 datasets. Specifically, we vary the pooling ratio from 0.2 to 0.8 with a step length of 0.2, the number of layers from one to four, and the hidden layer dimensions range from 16 to 128. The results are presented in Fig. 3. We can observe that BernPool overall exhibits robustness to variations of parameters. However, performance fluctuations can be observed on the NCI1 dataset when we evaluate the effect of different pooling ratios for BernPool. This may be attributed to the relatively less average number of nodes resulting in less information being retained after performing three pooling layers consecutively. The results shown in Fig. 3(b) indicate that setting the layer number to three achieves the best performance on the PROTEINS and NCI1 datasets. However, increasing the number of layers will require more computation resources and longer training time. As shown in Fig. 3(c), the highest accuracy is achieved when the dimension size is set as 128. With the dimension increasing, the accuracy presents a slight increase trend, which suggests that increasing the dimension size can enhance the model's capacity to capture more complex representations of the input graph. However, larger dimension sizes increase computation burdens. Thus we set the pooling ratio as 0.8, the layer number as 3, and the dimension as 128 in our framework.

## 7 Conclusion

In this paper, we proposed a simple and effective graph pooling method, called Graph Bernoulli Pooling (BernPool) to promote the graph classification task. Specifically, a probabilistic Bernoulli sampling was designed to estimate the sampling probabilities of graph nodes, and to further extract more useful information, we introduced a learnable reference set to encode nodes into a latent expressive probability space. Compared with the deterministic way, BernPool possessed more diversity to capture salient substructures. Then, to jointly learn representative substructures and preserve graph topology information, we proposed a hybrid graph pooling paradigm that fuses two pooling manners. We evaluate BernPool on multiple widely used datasets and dissected the framework with ablation analysis. The experimental results show that BernPool outperforms state-of-the-art methods and demonstrates the effectiveness of our proposed modules.

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
