# Supplementary material for "Graph Bernoulli Pooling"

## 1 Bernoulli Sampling Optimization Objective Derivation Details

**Notations.** $\phi$ denotes the parameter set of the BernPool module, $\psi$ is the parameter set of the other modules.

**Mutual Information Maximizing.** BernPool aims to maximize the mutual information between learned subgraph embeddings and corresponding labels, which can be formulated as:

$$\zeta_{MI} = MI(\mathbf{y}, f_{\psi,\phi}(\mathcal{G}, \mathcal{S})) = MI(\mathbf{y}, \mathbf{f}), \tag{1}$$

where $f_{\psi,\phi}$ represents the graph embedding process. Moreover, we introduce $\mathbf{f}$ to denote $f_{\psi,\phi}(\mathcal{G}, \mathcal{S})$ for simplification, which distributes in the embedding space $\mathcal{F}$. $\mathcal{F}$ is spanned by the resulted embeddings of $f_{\psi,\phi}$ inferred based on the input $\mathcal{G}$ and reference set $\mathcal{S}$. Then, based on the connection between the mutual information and entropy, the objective can be further written as:

$$\begin{aligned} &\arg\max MI(\mathbf{y}, \mathbf{f}) \\ &= \arg\max H(\mathbf{y}) - H(\mathbf{y}|\mathbf{f}), \end{aligned} \tag{2}$$

where $H(\mathbf{y})$ can be just omitted from the objective as it is independent from $\widetilde{\psi}, \widetilde{\phi}$. We have the following derivation:

$$\begin{aligned} &\arg\max - H(\mathbf{y}|\mathbf{f}) \\ &= \arg\max \sum_i -H(\mathbf{y}|\mathbf{f} = \mathcal{F}_i)p(\mathcal{F}_i) \\ &= \arg\max \sum_i p(\mathcal{F}_i)\mathbb{E}_{\mathbf{y}|\mathcal{F}_i}(\log p(\mathbf{y}|\mathbf{f} = \mathcal{F}_i)), \end{aligned} \tag{3}$$

where $p(\mathcal{F}_i)$ means the probability of the $i$-th observation in the embedding space and can be rationally assumed to conform to the uniform distribution. L denotes the number of observations. As the observation $\mathcal{F}_i$ means to be inferenced based on an input sample $\mathcal{G}_i$ with $\mathcal{S}$, we further denote $p(y|\mathbf{f} = \mathcal{F}_i)$ equally $p_{\psi,\phi}(\mathbf{y}|\mathcal{G}_i, \mathcal{S})$. The objective can be further written as:

$$\begin{aligned} &\arg\max \sum_i \frac{1}{L}\mathbb{E}_{\mathbf{y}|\mathcal{G}_i,\mathcal{S}}(\log p_{\psi,\phi}(\mathbf{y}|\mathbf{f} = \mathcal{F}_i)) \\ &= \arg\max \sum_i \mathbb{E}_{\mathbf{y}|\mathcal{G}_i,\mathcal{S}}[\log \int p_{\psi}(\mathbf{y}|\mathcal{G}_i, \mathcal{S}, \mathbf{z})p_{\phi}(\mathbf{z}|\mathcal{G}_i, \mathcal{S})d\mathbf{z}] \\ &= \arg\max \sum_i \mathbb{E}_{\mathbf{y}|\mathcal{G}_i,\mathcal{S}}[\log \int q_{\phi}(\mathbf{z}|\mathcal{G}_i, \mathcal{S})p_{\psi}(\mathbf{y}|\mathcal{G}_i, \mathcal{S}, \mathbf{z})\frac{p_{\phi}(\mathbf{z}|\mathcal{G}_i, \mathcal{S})}{q_{\phi}(\mathbf{z}|\mathcal{G}_i, \mathcal{S})}d\mathbf{z}], \end{aligned} \tag{4}$$

where $p_{\psi}(\mathbf{y}|\mathcal{G}_i, \mathcal{S}, \mathbf{z})$ is the conditional probability of label $\mathbf{y}$. $p_{\phi}(\mathbf{z}|\mathcal{G}_i, \mathcal{S})$ denotes the conditional probability of the factor $\mathbf{z}$, which is usually intractable. Hence, we resort to the variational inference to approximate the intractable true posterior with $q_{\phi}(\mathbf{z}|\mathcal{G}_i, \mathcal{S})$ that is the expected distribution. According

to the Jensen Inequality, the above formulation can be deduced as follows:

$$\geq \arg\max \sum_i \mathbb{E}_{\mathbf{y}|\mathcal{G}_i,\mathcal{S}}[\int q_\phi(\mathbf{z}|\mathcal{G}_i,\mathcal{S}) \log(p_\psi(\mathbf{y}|\mathcal{G}_i,\mathcal{S},\mathbf{z}) \frac{p_\phi(\mathbf{z}|\mathcal{G}_i,\mathcal{S})}{q_\phi(\mathbf{z}|\mathcal{G}_i,\mathcal{S})}) d\mathbf{z}]$$

$$= \arg\max \sum_i \mathbb{E}_{\mathbf{y}|\mathcal{G}_i,\mathcal{S}}[\int q_\phi(\mathbf{z}|\mathcal{G}_i,\mathcal{S}) \log p_\psi(\mathbf{y}|\mathcal{G}_i,\mathcal{S},\mathbf{z}) + q_\phi(\mathbf{z}|\mathcal{G}_i,\mathcal{S}) \log \frac{p_\phi(\mathbf{z}|\mathcal{G}_i,\mathcal{S})}{q_\phi(\mathbf{z}|\mathcal{G}_i,\mathcal{S})} d\mathbf{z}] \quad (5)$$

$$= \arg\max \sum_i \mathbb{E}_{\mathbf{y}|\mathcal{G}_i,\mathcal{S}}[\mathbb{E}_{q_\phi(\mathbf{z}|\mathcal{G}_i,\mathcal{S})}[\log p_\psi(\mathbf{y}|\mathcal{G}_i,\mathcal{S},\mathbf{z})]] - D_{KL}(q_\phi(\mathbf{z}|\mathcal{G}_i,\mathcal{S})||p_\phi(\mathbf{z}|\mathcal{G}_i,\mathcal{S})).$$

As $q_\phi(\mathbf{z}|\mathcal{G}_i,\mathcal{S})$ is predefined distribution, $\mathbb{E}_{q_\phi(\mathbf{z}|\mathcal{G}_i,\mathcal{S})}$ can be regarded as a constant, the objective can be formulated as:

$$\arg\max \mathbb{E}_{\mathbf{y}|\mathcal{G},\mathcal{S}}[\log p_\psi(\mathbf{y}|\mathcal{G}_i,\mathcal{S},\mathbf{z})] - D_{KL}(q_\phi(\mathbf{z}|\mathcal{G}_i,\mathcal{S})||p_\phi(\mathbf{z}|\mathcal{G}_i,\mathcal{S}))$$

$$= \arg\max -\zeta_{CE} - D_{KL}(q_\phi(\mathbf{z}|\mathcal{G}_i,\mathcal{S})||p_\phi(\mathbf{z}|\mathcal{G}_i,\mathcal{S})) \quad (6)$$

$$= \arg\min \zeta_{CE} + D_{KL}(q_\phi(\mathbf{z}|\mathcal{G}_i,\mathcal{S})||p_\phi(\mathbf{z}|\mathcal{G}_i,\mathcal{S})).$$

In addition, we can extend the above single-layer BernPool into multi-layer networks by deploying independent sampling factors in sequential graph pooling.

**Cross-entropy Loss Function $\zeta_{CE}$ based on Subgraph Sampling.** Referring to the analysis of a random dropping method [1], we analyze the loss function of our proposed BernPool. We can derive two parts from $\zeta_{CE}$:

$$\zeta_{CE} = \mathcal{L}_{CE} + \sum_i \frac{1}{2} y_i(1-y_i) Var(\widetilde{h}_i), \quad (7)$$

where $\mathcal{L}_{CE}$ is the original cross-entropy loss function, the second term tends the classification probability to 0 or 1 and reduces the variance of $h_i$ in the training process.

Specifically, for analytical simplicity, we apply a single-layer graph convolution as the backbone model to perform the binary classification task. As mentioned in Section 3 of this paper, $\mathbf{H} = \sigma(\hat{\mathbf{D}}^{-\frac{1}{2}}\hat{\mathbf{A}}\hat{\mathbf{D}}^{\frac{1}{2}}\mathbf{XW})$ and $\mathbf{y} = \text{sigmoid}(\mathbf{H})$ represents predicted probability. Thus the original cross-entropy loss function can be expressed as follows:

$$\mathcal{L}_{CE} = \sum_{j,y_j=1} \log(1+e^{-h_j}) + \sum_{k,y_k=0} \log(1+e^{h_k}). \quad (8)$$

When performing sampling in the original graph, the objective function can be regarded as adding a bias, which is expressed as follows:

$$E(\zeta_{CE}) = \sum_{j,y_j=1} [\log(1+e^{-h_j}) + \mathbb{E}(u(\widetilde{h}_j,h_j))] + \sum_{k,y_k=0} [\log(1+e^{h_k}) + \mathbb{E}(v(\widetilde{h}_k,h_k))]. \quad (9)$$

$$\begin{cases} u(\widetilde{h}_j,h_j) = \log(1+e^{-\widetilde{h}_j}) - \log(1+e^{-h_j}). \\ v(\widetilde{h}_k,h_k) = \log(1+e^{-\widetilde{h}_k}) - \log(1+e^{-h_k}). \end{cases} \quad (10)$$

We can approximate it with second-order Taylor expansion of $u(\cdot)$ and $v(\cdot)$ around $h_j$ and $h_k$, respectively. For instance:

$$u(\widetilde{h}_j,h_j) = \frac{-e^{-h_j}}{1+e^{-h_j}}(\widetilde{h}_j - h_j) + \frac{1}{2}\frac{e^{-h_j}}{(1+e^{-h_j})^2}(\widetilde{h}_j - h_j)^2$$

$$= (-1+y_j)(\widetilde{h}_j - h_j) + \frac{1}{2}y_j(1-y_j)(\widetilde{h}_j - h_j)^2. \quad (11)$$

In the same way, $v(\widetilde{h}_k,h_k) = y_k(\widetilde{h}_k - h_k) + \frac{1}{2}y_k(1-y_k)(\widetilde{h}_k - h_k)^2$. So the above equation can be transformed as:

$$E(\zeta_{CE}) = \mathcal{L}_{CE} + E(\sum_{j,y_j=1} [(-1+z_j)(\widetilde{h}_j - h_j) + \frac{1}{2}y_j(1-y_j)(\widetilde{h}_j - h_j)^2])$$

$$+ E(\sum_{k,y_k=1} [z_k(\widetilde{h}_k - h_k) + \frac{1}{2}y_k(1-y_k)(\widetilde{h}_k - h_k)^2]) \quad (12)$$

$$= \mathcal{L}_{CE} + \sum_i \frac{1}{2}y_i(1-y_i)Var(\widetilde{h}_i).$$

# References

[1] Taoran Fang, Zhiqing Xiao, Chunping Wang, Jiarong Xu, Xuan Yang, and Yang Yang. Dropmessage: Unifying random dropping for graph neural networks, 2023.