# OpenReview forum: "Graph Bernoulli Pooling"
_NeurIPS.cc/2023/Conference — Submitted to NeurIPS 2023_

### Official Review · Reviewer_Zzuw · 2023-06-08

**Soundness:** 1 poor
**Presentation:** 3 good
**Contribution:** 1 poor
**Rating:** 3
**Confidence:** 5

**Summary:**

This paper studies the problem of graph classification and proposes a method named graph Bernoulli pooling (BernPool).  Considering the complementarity of node dropping and node clustering, the paper proposes a hybrid graph pooling paradigm to combine a compact subgraph (via dropping) and a coarsening graph (via clustering), in order to retain both representative substructures and input graph info.

**Strengths:**

1. The problem is important.

2. The writing is good.

**Weaknesses:**

1. The results are weirdly high. I checked the code and found no validation set. First, the experimental setting is different from MVGPool, but author directly copies the number. Second, for example, authors select the worst model variant of MVGPool in the original paper.

2. Node dropping and clustering are both well-known strategies for graph pooling. Combining them is not novel to this community.

3. The ablation studies show your both variants highly outperform the best baseline on ENZYME, for example. So, which part works most in your framework, not the clustering or dropping? Can you explain more?


**Questions:**

see weaknesses.

**Limitations:**

see weaknesses.

---

> ### Author Rebuttal · Authors · 2023-08-09
>
> We sincerely thank the Reviewer's comments.
>
> **Q1.** The results are weirdly high. I checked the code and found no validation set. First, the experimental setting is different from MVGPool, but author directly copies the number. Second, for example, authors select the worst model variant of MVGPool in the original paper.
>
> **A1.** For the concern about no validation set, we would like to clarify that we **strictly** follow the **standard** and **popular** 10-fold cross-validation protocol (widely adopted in DiffPool [NeurIPS 2020], VIPool [NeurIPS 2020], MPool [KDD 2023], SEP-G [PMLR 2022], SAGPool [PMLR 2019], ASAP [AAAI 2020], GSAPool [WWW 2020], iPool [TNNLS 2021]) for the graph classification task for a fair comparison.
>
> For your concern about the experimental setting different from MVPool, here we additionally follow the **standard** 10-fold cross-validation protocol to run the variant "MVPool_SL" which has the best performance in [MVPool, TKDE 2021], using the MVPool-released official code. The results are shown in below, and our BernPool still outperforms "MVPool_SL" on seven datasets (OOM of MVPool_SL on the DD dataset).   We have uploaded the code of "MVPool_SL"  that employs the standard 10-fold cross-validation protocol for evaluating in the link of the Abstract. Besides, we will update the results of "MVPool" in Table 1 in the next verison.
>
> |  Method   |    PROTEINS    |       DD       |      NCI1      |  Mutagenicity  |    ENZYMES     |     IDMB-B     |     IMDB-M     |     COLLAB     |
> | :-------: | :------------: | :------------: | :------------: | :------------: | :------------: | :------------: | :------------: | :------------: |
> | MVPool_SL |   81.67±2.73   |       -        |   81.22±1.16   |   83.12±1.60   |   62.50±4.73   |   80.60±3.47   |   55.87±4.70   |   84.56±1.08   |
> | BernPool  | **83.29±3.69** | **83.27±2.95** | **81.44±1.09** | **83.81±1.43** | **76.00±3.78** | **81.30±3.50** | **55.93±3.80** | **85.26±1.35** |
>
> **Q2.** Node dropping and clustering are both well-known strategies for graph pooling. Combining them is not novel to this community.
>
> **A2.** From a high-level cognition, indeed, most graph pooling methods abide by node dropping and clustering, but this cannot simply deny the novelty or contribution of our method. For our method, because the proposed "Bernoulli node dropping" and "Bernoulli node clustering" are novel algorithms, the fusion of them has novelty and contribution. Specifically, different from the existing node dropping and clustering algorithms, our "Bernoulli node dropping" and "Bernoulli node clustering" both employ a variational Bernoulli learning process to estimate the sampling probabilities of graph nodes. To restrict the sampling process, we formulate a variational Bernoulli learning constraint by deriving an upper bound between an expected distribution and a learned distribution. In particular, "Bernoulli node clustering" is also Bernoulli-induced and more efficient than the previous clustering methods because the learning of the assignment matrix is bypassed.
>
> **Q3.** The ablation studies show both variants highly outperform the best baseline on ENZYMES, for example. So, which part works most in your framework, not the clustering or dropping? Can you explain more?
>
> **A3.** You should have some misunderstanding about the comparison in ablation studies of Table 3 in the manuscript. “BernPool w/o Clustering” contains the Bernoulli sampling factor learning and dropping, and “BernPool w/o Dropping” contains the Bernoulli sampling factor learning and clustering. It means that the Bernoulli sampling factor learning must be involved, otherwise, graph pooling cannot be performed. Therefore, our designed Bernoulli sampling plays an important role in improving performance, please see Table 3 and Table 4 in the manuscript.

---

> > ### Comment · Reviewer_Zzuw · 2023-08-19
> > **Thanks your response.**
> >
> > Thanks your response. I thought combine them is your novelty since learning important nodes for graphs in a probabilistic way is also not new [1]. The experiments are still less convinced, and I suggest to add more visualization or formal theoretical analysis given the method is kind of simple.
> >
> > [1] Learning Graph Augmentations to Learn Graph Representations

---

> > > ### Author Response · Authors · 2023-08-20
> > > **Reclarification of experiment results, novelty, theoretical anaysis and visualizations**
> > >
> > > We sincerely thank your response.
> > >
> > > For the question about “the experiments are still less convinced”, we must reclarify: i) we **strictly follow the standard protocols** on all used datasets for fair comparisons; ii) **10 widely-used datasets are evaluated**, and the experiment results have demonstrated the effectiveness; iii) **the code has been released** for reproducibility. You can **directly verify** the experiment results on the used ten datasets.
> > >
> > > For your concern about novelty, besides the combination of them, the variational inference-based Bernoulli sampling is one of our contributions. For the literature [1] you argued, we clarify the methodological difference from ours.
> > >
> > > The work [1] aimed to augment graph samples by adding random disturbances on edges/nodes/features based on the Gumbel-Softmax trick [a] [b]. In principle, this work [1] should fall into the category of graph augmentation, instead of graph pooling for ours. Moreover, our method derives a bound of variational inference to control the sampling proportion for pooling, while the work [1] simply used random sampling on Bernoulli distribution for augmentation.
> > >
> > > For the visualization, in fact, we have provided some examples in the original rebuttal file (see the PDF file). For more examples, we have uploaded them to the folder “Visualizations” in the anonymous link of ABSTRACT, please see the visualizations (“xx_visualization.png”) and corresponding data (“xx_data. pkl”).
> > >
> > > For your concern about theoretical analysis, we actually provided the theoretical analysis of stability in the supplementary material (please see Section “Cross-entropy Loss Function …”). Below we give a brief analysis again.
> > >
> > > First, the original cross-entropy loss function can be expressed:
> > > $$
> > > \mathcal{L}\_{CE} = \sum\_{j,y\_{j}=1} \log(1 + e^{-h\_j}) + \sum\_{k,y\_{k}=0} \log(1 + e^{h\_k}).
> > > $$
> > > When performing sampling in the original graph, the above objective function can be regarded as adding a bias, which is expressed:
> > > $$
> > > E(\zeta\_{CE}) = \sum\_{j,y\_{j}=1} [\log(1 + e^{-h\_j}) + \mathbb{E}(u(\widetilde{h}\_j, h\_j))] + \sum\_{k,y\_{k}=0} [\log(1 + e^{h\_k}) + \mathbb{E}(v(\widetilde{h}\_k,h\_k))].
> > > $$
> > > Then, we can approximate the above equation with second-order Taylor expansion of $u(\cdot)$ and $v(\cdot)$ around $h\_j$ and $h\_k$, respectively. So the above equation can be transformed as:
> > > $$
> > > \mathcal{L}\_{CE} + \sum\_i \frac{1}{2} y\_i (1 - y\_i) Var(\widetilde{h}\_i).
> > > $$
> > > According to the above equation, the variance of the learned graph feature is involved as a specific variance regularization. Specifically, sample variance measures the degree of stability [c]. By reducing the expectation of loss function through training, the sample variance is accordingly reduced. Hence, BernPool contributes to stabilizing the training and inference process. The detailed derivation can be found in the supplementary material.
> > >
> > > [a] Jang E, Gu S, Poole B. Categorical reparameterization with gumbel-softmax[J]. arXiv preprint arXiv:1611.01144, 2016.
> > >
> > > [b] Maddison C J, Mnih A, Teh Y W. The concrete distribution: A continuous relaxation of discrete random variables[J]. arXiv preprint arXiv:1611.00712, 2016.
> > >
> > > [c] Fang T, Xiao Z, Wang C, et al. Dropmessage: Unifying random dropping for graph neural networks[C]//Proceedings of the AAAI Conference on Artificial Intelligence. 2023, 37(4): 4267-4275.

---

### Official Review · Reviewer_rF61 · 2023-07-04

**Soundness:** 2 fair
**Presentation:** 3 good
**Contribution:** 2 fair
**Rating:** 5
**Confidence:** 4

**Summary:**

Graph pooling is an essential operator in GNNs for graph classification. This paper propose a hierarchical pooling method which is based on the node clustering. Existing graph pooling methods are designed in a deterministic way which neglect the intrinsic structure feature. This paper propose BernPool, which first conducts Bernoulli sampling and then combines a compact subgraph after node dropping and a coarsening graph after node clustering. BernPool can jointly learn expressive substructures and keep graph topology. The core of this paper is the utilization of Bernoulli sampling factor for node selection and clustering.

**Strengths:**

(1)	The topic of graph pooling is important in graph representation learning.

(2)	The writing of this paper is clear.

**Weaknesses:**

(1)	The model's complexity does not effectively demonstrate the necessity or unique advantages of each intricate design choice. The model first adopts a complex strategy for sampling, and then employs a complex hybrid pooling strategy, which lacks significant insights into their effectiveness.

(2)	Insufficient experimentation. The dataset used in the experiments is too small, and the effectiveness should be validated on larger datasets, such as ogb and zinc.

(3)	The method has excessively high complexity, which can limit its scalability in practical applications.

**Questions:**

Does the proposed method have higher expressive power?

**Limitations:**

yes

---

> ### Author Rebuttal · Authors · 2023-08-09
>
> We sincerely thank the reviewer's comments.
>
> **Q1.** The model’s complexity does not effectively demonstrate the necessity or unique advantages of each intricate design choice. The model first adopts a complex strategy for sampling, and then employs a pooling strategy, which lacks significant insights into their effectiveness.
>
> **A1.** First, we clarify that the proposed sampling strategy is very simple and effective, both in the technique line and the experimental comparisons. Besides, the implementation code only has a few lines and could be plug-and-play in those GNNs, please see the released code. For quantitative analysis of efficiency, we provide the complexity comparison with other pooling methods in Table D. Also, we additionally show the comparisons of average training time and inference time of one epoch, and memory usage in the DD dataset in Table C.
>
> Second, for the questions about significant insights into the effectiveness, we make the explanation as follows: Compared with the deterministic algorithms, the Bernoulli sampling has better data-adaptability, and exploration ability and achieves better performance. Specifically, 1) Adaptability to data variability: BernPool-sampling nodes in a probabilistic manner are based on the computed node importance, which reflects the significance of each node in the graph. Also, BernPool is able to produce meaningful data-dependent probabilistic measures of uncertainty [a]. 2) Exploration: The stochastic nature of Bernoulli sampling introduces an exploratory aspect, enabling the model to explore different sub-structures during pooling. We visualize the exploration process in Fig. A. Based on the sampling probability p, the multiple sampling situations of z are inferred. This exploration improves the representation of the underlying data distribution. 3) According to the experiment results in Table 1 of the manuscript, our BernPool achieves better performance than the deterministic pooling methods. Besides, we have fully analyzed the effectiveness of our BernPool in Table 2, Table 3, and Table 4 of the manuscript. The experiment results demonstrate that our BernPool is plug-and-play and can effectively promote performance. More details can be found in Section 6.
>
> [a] Martin R, Liu C. Inferential models: A framework for prior-free posterior probabilistic inference[J]. Journal of the American Statistical  Association, 2013, 108(501): 301-313.
>
>
>
> **Q2.** Insufficient experimentation. The dataset used in the experiments is too small, and the effectiveness should be validated on larger datasets, such as ogb and zinc.
>
> **A2.** The employed benchmark datasets are widely adopted and popular in previous works. Actually, the number of average nodes in the DD dataset is more than in the ogb dataset.
>
> For the larger datasets you concerned, we additionally conduct experiments on the ogbg-ppa (for graph classification) and do not use “zinc” because it is used for the regression task, different from the focus of our work. Instead, we test on another larger dataset: REDDIT-MULTI-12K. The experiment results on both datasets are reported in Table A and Table B. Please note that, for the ogbg-ppa dataset, we strictly followed the standard protocol released by OGB officials. According to the results, BernPool outperforms other pooling methods, demonstrating BernPool’s effectiveness and superiority on large datasets.
>
> **Q3.** The method has excessively high complexity, which can limit its scalability in practical applications.
>
> **A3.** We clarify that the proposed method is very simple, and has a low computation complexity. For quantitative analysis of efficiency, we provide the complexity comparison with other pooling methods in Table D. Also, we additionally show the comparisons of average training time and inference time of one epoch, and memory usage in the DD dataset in Table C. At the same time, the implementation code only has a few lines, please see the released code.
>
> For scalability in practical applications, our BernPool is plug-and-play (Please see Table 2 in the manuscript) and can also achieve high performance on large datasets. More details can be seen in answers to **Q1** and **Q2**.
>
> **Q4.** Does the proposed method have higher expressive power?
>
> **A4.** For your concern about expressive power, we employ the t-SNE to visualize the hidden layer feature after three layers of MVPool and BernPool on the PROTEINS dataset in Fig. B, respectively. According to the feature visualization, the pooled features after BernPool present better inter-class dispersion and intra-class compactness than MVPool, demonstrating the expressive power of our method. Besides, according to experimental results, our BernPool outperforms the existing state-of-the-art pooling methods, demonstrating the better representation learning ability of our BernPool.

---

> > ### Comment · Reviewer_rF61 · 2023-08-19
> > **Official Comment by Reviewer rF61**
> >
> > Thank you very much for the careful response and the addition of new experiments. I have increased my score by 1 point due to the effort you have put in.
> >
> > However, I do not actually agree with this route. Many pooling methods based on complex sampling or clustering mechanisms have been proposed, and their complexity even exceeds that of some high expressive GNNs, such as subgraph GNN methods. But the benefits of these methods in the downstream tasks are much lower compared to high expressive GNNs.
> >
> > Please let the AC make the final decision.

---

### Official Review · Reviewer_kSdL · 2023-07-05

**Soundness:** 3 good
**Presentation:** 2 fair
**Contribution:** 2 fair
**Rating:** 6
**Confidence:** 3

**Summary:**

This work proposes a probabilistic hierarchical graph pooling method called graph Bernoulli pooling (BernPool) for graph features learning.
Unlike previous deterministic works, the probabilistic Bernoulli sampling with the reparameterization trick allows BernPool to preserve certain diversity during the downsampling without losing high efficiency.
Besides, BernPool can capture representative substructures of graphs by combining the advantage of node-dropping and node-clustering techniques.
Ample empirical experiments demonstrate the effectiveness of the proposed methods, which outperform previous works with remarkable gaps and fewer parameters.
The ablation study justifies each proposed component in BernPool.




**Strengths:**

1. This work proposes new a hierarchical pooling method combining node-clustering and node-dropping and introduces probabilistic properties with a Bernoulli sampling framework. The Bernoulli optimization objective with soft-orthogonal constraint also contributes to performance improvement.
2. The proposed method outperforms previous techniques with remarkable performance gaps. Besides, the method is also applicable to several typical GNN backbones and shows decent improvements.
3. The proposed method is well supported by comprehensive empirical comparisons and ablation studies.
4. The mathematical derivation supports the proposed Bernoulli sampling well theoretically.

**Weaknesses:**

1. There are some minor confusing points in the paper writing. (listed in the question)
2. The authors explain the motivation for using probabilistic techniques in selecting reference sets and mention the drawbacks of node-dropping pooling methods.
However, the motivation for the hybrid graph pooling is vague. The authors categorize BernPool as a node clustering pooling method but do not explain the motivation for adding back node dropping counterpart.


**Questions:**

1. I am a bit confused about the reparameterization trick mentioned in line 170 for Bernoulli distribution. I assume it shall be something similar to [1]. If so, I think you should cite it. If I am wrong or missing something, please provide more details.
2. What is the motivation for adding node-dropping, and why it helps?
According to the ablation study, removing node-dropping will lead to a slight performance drop except on NCI1 and ENZYMES, on which BernPool is impacted notably.
3. How BernPool works during the inference? Especially regarding the Bernoulli sampling. Will BernPool have multiple inferences with different sampling, similar to MC-dropout [2] and the reference [21] listed in the paper?
4. When introducing multiple BernPool layers in a neural network,  $\mathbf{H}$ in eq. (7) will be the same for all BernPool layers, or it refers to the node representations of the coarsened graphs at the corresponding layer?



------------
- [1] Bengio, Yoshua, Nicholas Léonard, and Aaron Courville. "Estimating or propagating gradients through stochastic neurons for conditional computation." arXiv preprint arXiv:1308.3432 (2013).
- [2] Gal, Yarin, and Zoubin Ghahramani. "Dropout as a Bayesian approximation: Representing model uncertainty in deep learning." international conference on machine learning. PMLR, 2016.

---

> ### Author Rebuttal · Authors · 2023-08-09
>
> We sincerely thank you for recognizing the positive aspects of our paper, such as the novelty, the extensiveness of experiments, and the competitive performance.
>
> **Q1.** I am a bit confused about the reparameterization trick mentioned in line 170 for Bernoulli distribution. I assume it shall be something similar to [1]. If so, I think you should cite it. If I am wrong or missing something, please provide more details.
>
> **A1.** Our employed reparameterization trick is motivated by [a], and meantime different from [1]. The reparameterization trick can be used for efficient approximate posterior probability with continuous latent variables and/or parameters by deriving a variational lower bound. And it is straightforward to optimize using standard stochastic gradient ascent techniques [a]. Different from [a] in which the approximate posterior (Gaussian) is related to the input data, the approximate posterior (Bernoulli) of our BernPool is derived from both input data and the reference set. On the other hand, [1] uses the Bernoulli distribution to generate stochastic binary units, but does not involve estimating posterior probabilities by deriving variational lower bounds and even no Bayesian inference. Strictly speaking, [1] does not conform to the idea of reparameterization proposed in [a].
>
> [a] Kingma D P, Welling M. Auto-encoding variational bayes[J]. arXiv preprint arXiv:1312.6114, 2013.
>
> **Q2.** What is the motivation for adding node-dropping, and why it helps? According to the ablation study, removing node-dropping will lead to a slight performance drop except on NCI1 and ENZYMES, on which BernPool is impacted notably.
>
> **A2.**  In one graph, there may exist redundant nodes which may degrade classification performance. The purpose of node-dropping is to adaptively learn the importance of graph nodes and select a part of salient nodes by conforming to the expected distribution.
>
> According to Table 3 in the manuscript, our “Bernoulli node dropping” promotes the classification of the Bioinformatics datasets with a performance gain from 0.9 to 5.2. This demonstrates the effectiveness of “Bernoulli node dropping”. Compared with the Bioinformatics datasets, in the social network datasets, the less improvement may be attributed to the higher edge-node ratios of social network datasets than those of Bioinformatics datasets, which may indicate less node redundancy.
>
> **Q3.** How BernPool works during the inference? Especially regarding the Bernoulli sampling. Will BernPool have multiple inferences with different sampling, similar to MC-dropout [2] and the reference [2] listed in the paper?
>
> **A3.** Similar but different to the reference [2], BernPool introduces a learnable _probability_ sampling while [2] used a random drop-out, although both employ the widely-used strategy: multiple inferences. Below we reclarify the inference of our BernPool:
> During the inference, the preserved nodes after pooling are obtained by sampling from the learned Bernoulli distribution. The visualization of the sampling process in Fig. A, also demonstrates the sampling variations. Nevertheless, after training with probabilistic learning, our model can be robust to the sampling variations conforming to the learned distribution. To verify the robustness, we run ten times for a given sample and the variance of probabilities of predicting the correct class is 0.0096. The results demonstrate the stability of classification.
>
> **Q4.** When introducing multiple BernPool layers in a neural network, **H** in eq. (7) will be the same for all BernPool layers, or it refers to the node representations of the coarsened graphs at the corresponding layer?
>
> **A4.** The symbol H in eq. (7) refers to the node representation of the coarsened graphs at the corresponding layer.
>
> **Q5.** The authors explain the motivation for using probabilistic techniques in selecting reference sets and mention the drawbacks of node-dropping pooling methods. However, the motivation for the hybrid graph pooling is vague. The authors categorize BernPool as a node clustering pooling method but do not explain the motivation for adding back node dropping counterpart.
>
> **A5.** The reason for adding back node dropping is that the “Bernoulli node dropping” and “Bernoulli node clustering” are actually complementary. “Bernoulli node clustering” can preserve the local information but may lead to node smooth through node aggregation. In contrast, “Bernoulli node dropping” can preserve the saliency of representative nodes while inevitably missing some structural information. Hence, considering the large structure variations among graph data, fusing them would be beneficial for graph representation.

---

> > ### Comment · Reviewer_kSdL · 2023-08-17
> >
> > Thank you for the responses.
> >
> > I will raise my score to 6.

---

### Official Review · Reviewer_PxAC · 2023-07-06

**Soundness:** 3 good
**Presentation:** 4 excellent
**Contribution:** 3 good
**Rating:** 6
**Confidence:** 5

**Summary:**

This paper proposes a pooling method that probabilistically generates a coarse graph using an expected sampling strategy. A hybrid pooling paradigm is designed to integrate two existing pooling paradigms. Extensive experiments demonstrate the effectiveness of the proposed method.


**Strengths:**

The method is well-organized and easy to follow.

The proposed approach is simple yet effective, as supported by the experimental results.

**Weaknesses:**

The stability of the proposed method is unclear. Since different runs may result in different factors z, which in turn lead to different projection matrices P and coarse graphs, the stability of the method needs to be considered. It would be important to address this issue, as obtaining different results in each inference procedure can be problematic.

In the experiments, it would be beneficial to provide examples that illustrate the reference set S, probability P, and the factor z. Including these examples would enhance the readers' understanding of the proposed method and its components.


**Questions:**

Please check the weaknesses.

**Limitations:**

Please check the weaknesses.

---

> ### Author Rebuttal · Authors · 2023-08-09
>
> We sincerely thank the reviewer's comments.
>
> **Q1.** The stability of the proposed method is unclear. Since different runs may result in different factors z, which in turn lead to different projection matrices P and coarse graphs, the stability of the method needs to be considered. It would be important to address this issue, as obtaining different results in each inference procedure can be problematic.
>
>
>
> **A1.** During the inference, the preserved nodes after pooling are obtained by sampling from the learned Bernoulli distribution. Hence, BernPool indeed has multiple inferences with different sampling. The visualization of the sampling process in Fig. A also demonstrates the sampling variations. Nevertheless, after training with probabilistic learning, our model can be robust to the sampling variations conforming to the learned distribution. _To verify the robustness, experimentally, we run ten times for a given sample and the variance of probabilities of predicting the correct class is 0.0096._ The results demonstrate the stability of classification.
>
> _At the same time, in the supplementary material, theoretically, we have analyzed the objective of BernPool and the loss function based on the BernPool sampling. By reducing the sample variance that is used to measure the degree of stability [a], BernPool contributes to stabilizing the training and inference process._ As demonstrated in Eqn. (12) of the supplementary material, BernPool introduces a specific variance regularization to the objective function to reduce the sample variance through training.
>
> [a] Fang T, Xiao Z, Wang C, et al. Dropmessage: Unifying random dropping for graph neural networks[C]//Proceedings of the AAAI Conference on Artificial Intelligence. 2023, 37(4): 4267-4275.
>
>
>
> **Q2.** In the experiments, it would be beneficial to provide examples that illustrate the reference set S, probability P, and the factor z. Including these examples would enhance the readers' understanding of the proposed method and its components.
>
>
>
> **A2.** We additionally provided visualization examples. Specifically, we visualize the reference set with the orthogonal constraint in Fig. A (e). To show the reference set's orthogonality, we visualize the cosine similarity of the reference set in Fig. A (f). Also, we show the example of probability vector p by providing the value of each element in the table of Fig. A. Based on the sampling probability p, the three sampling situations of z-1, z-2, and z-3 (also corresponding to sub-figures (b), (c) and (d)) are shown in the table of Fig. A.

---

> > ### Comment · Reviewer_PxAC · 2023-08-20
> >
> > I have read the response carefully, and all my questions can be addressed. I will keep my score unchanged since the main contribution of this paper is unchanged in the rebuttal period.

---

### Official Review · Reviewer_rS8P · 2023-07-07

**Soundness:** 3 good
**Presentation:** 3 good
**Contribution:** 2 fair
**Rating:** 5
**Confidence:** 4

**Summary:**

The authors introduce a non-deterministic graph pooling method called graph Bernoulli pooling (BernPool). This work comprises two main components: a probabilistic Bernoulli sampling method that allows for more diverse sampling situations and captures intrinsic characteristics of the data, and a hybrid graph pooling paradigm that combines a compact subgraph with a coarsened graph. Experimental results demonstrate the significant superiority of BernPool over the baseline methods in graph classification tasks.

**Strengths:**

This work is easy to follow.
The idea presented in this paper is straightforward and the hybrid graph pooling technique significantly improves the performance of graph pooling.
The authors have provided code implementation, enhancing reproducibility and adding to the persuasiveness of their work.

**Weaknesses:**

1. More visualizations are needed to illustrate how node dropping and node clustering work, such as which nodes are discarded during the pooling process and which nodes are grouped into the same clusters.
2. Transforming the deterministic process of pooling into sampling has been explored in previous works [1]. Although BernPool presents a more refined approach, the authors provide insufficient discussion or theoretical analysis of why Bernoulli sampling is superior to deterministic algorithms.
[1] Liu, Ning, et al. "Unsupervised Hierarchical Graph Pooling via Substructure-Sensitive Mutual Information Maximization."


**Questions:**

Q1. Regarding the inference time, I guess that in line 301, when the authors mention the number of nodes and edges in the "proteins" dataset, it refers to the total number of nodes and edges across all graphs in the dataset (20,000/70,000). However, it is worth noting that the average number of nodes and edges of the graphs in the "proteins" dataset is relatively low (approximately 30/70). This might give readers a misconception that BernPool can effectively handle large graphs with tens of thousands of nodes. However, it is important to consider that the scalability of node clustering pooling often not be as efficient as node dropping pooling. The use of dense adjacency matrices in node clustering pooling can result in a higher space complexity of O(n^2), which limits its ability to handle large graphs. To provide further clarity on the contribution of BernPool, the authors could consider running experiments on datasets with a larger number of nodes (e.g., D&D dataset) and provide information on the memory usage and running time of BernPool compared to other baseline methods (e.g., DiffPool, MinCutPool, SAGPool, Topkpool). This would help in better understanding the capabilities and limitations of BernPool.

**Limitations:**

The proposed method appears to be somewhat incremental.

---

> ### Author Rebuttal · Authors · 2023-08-09
>
> We sincerely thank the reviewer's comments.
>
> **Q1.** More visualizations are needed to illustrate how node dropping and node clustering work, such as which nodes are discarded during the pooling process and which nodes are grouped into the same clusters.
>
> **A1.** We additionally provide visualization examples in Fig. A (b), (c), and (d). The preserved nodes of “Bernoulli node dropping” are also the clustering centers of “Bernoulli node clustering”. The aggregation process of clustering centers of “Bernoulli node clustering” is remarked in RED lines.
>
> **Q2.** Transforming the deterministic process of pooling into sampling has been explored in previous works [1]. Although BernPool presents a more refined approach, the authors provide insufficient discussion or theoretical analysis of why Bernoulli sampling is superior to deterministic algorithms. [1] Liu, Ning, et al. "Unsupervised Hierarchical Graph Pooling via Substructure-Sensitive Mutual Information Maximization."
>
> **A2.** Theoretically, our BernPool is completely different from the method [1]. The sampling probability distribution S of [1] is based on the data-independent uniform distribution, which can be regarded as adding random disturbances to the assignment matrix learning process. In contrast, our sampling distribution is data-dependent and obtained through variational inference.
>
> Compared with the deterministic algorithms, the Bernoulli sampling has better data-adaptability, and exploration ability and achieves better performance. Specifically, 1) Adaptability to data variability: BernPool sampling nodes in a probabilistic manner are based on the computed node importance, which reflects the significance of each node in the graph. Also, BernPool is able to produce meaningful data-dependent probabilistic measures of uncertainty [a]. 2) Exploration: The stochastic nature of Bernoulli sampling introduces an exploratory aspect, enabling the model to explore different sub-structures during pooling. We visualize this exploration process in the table of Fig. A. Based on the sampling probability p, the multiple sampling situations of z are inferred. This exploration improves the representation of the underlying data distribution. 3) According to the experiment results, our BernPool achieves better performance than the deterministic pooling methods.
>
> [a]. Martin R, Liu C. Inferential models: A framework for prior-free posterior probabilistic inference[J]. Journal of the American Statistical Association, 2013, 108(501): 301-313.
>
> **Q3.** i) Regarding the inference time, ... mention the number of nodes and edges in the "proteins" dataset ... However, ... the average number of nodes and edges of the graphs ... is relatively low (approximately 30/70). This might give readers ... handle large graphs ...
>
> ii) However, it is important to consider that the scalability ... The use of dense ... result in a higher space ..., which limits its ability to handle large graphs.
>
> iii) ... clarity on the contribution of BernPool, ... consider running experiments on ... (e.g., D&D dataset) and ... memory usage and running time ... compared to ... (e.g., DiffPool, MinCutPool, SAGPool, Topkpool). ... better understanding ... of BernPool.
>
> **A3.** i) For your concern about the misunderstanding of “20000 nodes …”, we will revise the sentence as “Thus we conduct experiments on the PROTEINS dataset to compare the inference time (single-layer).” for clarity.
>
> ii) For your concern about the “space complexity of O(n^2)”, in graph representation learning, the storage space depends on the used data structure. If using the general matrix, it is indeed O(n^2), which might limit its ability to handle large graphs. But in practice, as graphs is sparse (the number of edges is much less than n^2), we may only store paired node indices and the associated weight for those edges, and then operate this type of data structure. Besides, in our current work, we do not focus on such a problem.
>
> iii) For the large dataset DD you recommended, in fact, we had reported the Results of DD in Table 1 of the manuscript. For more verification for our method, we additionally conduct experiments on other two large datasets: ogbg-ppa and REDDIT-MULTI-12K, and the results are shown in Table A and Table B. According to the results, BernPool still achieves the best performance. The corresponding codes have been additionally uploaded to the link of Abstract.
>
> Besides, we compare the memory usage, average training time and inference time of one epoch between BernPool and other baseline methods on the DD dataset in Table C. The number of network layers is all set as 3, and the hidden layer dimension is set as 128. According to the results, our BernPool achieves the highest performance while taking less running time and medium memory consumption.

---

> ### Comment · Reviewer_rS8P · 2023-08-19
>
> I have looked into the response provided by the authors.
>
> Most of my concerns are well addressed. But Fig (a), (b) and (c) are more like schematic diagrams. I would recommend utilizing some **real pooling results**, as we found that in most of the pooling methods in GNNs, the behavior of pooling at the run time is quite different from what is drawn in the manuscript. This could my pretty misleading.
>
> I am still inclined to accept this paper and will keep my rating.

---

> > ### Author Response · Authors · 2023-08-20
> > **Visualizations of real pooling results**
> >
> > We sincerely thank your response. Following your suggestions, we have additionally visualized the **real pooling results** of some samples in PROTEINS, Mutagenicity, ENZYMES, IMDB-BINARY, and IMDB-MULTI datasets. The visualizations (“xx_visualization.png”) and corresponding data (“xx_data. pkl”) have been uploaded to the folder “Visualizations” in the anonymous link of ABSTRACT.

---

### Author Rebuttal · Authors · 2023-08-10

We sincerely thank the Area Chair and the reviewers for your efforts and valuable comments.

According to the comments, we additionally conduct multiple experiments and provide visualization examples to evaluate the performance and facilitate the understanding of our method. Specifically, i) We additionally conduct experiments on other large datasets (the ogbg-ppa and REDDIT-MULTI-12K datasets) to demonstrate the learning ability of our method. The results show in Table A and Table B. ii) We provide the Bernoulli sampling visualizations and feature visualization in Fig. A and Fig. B for an intuitive understanding. iii) We compare the memory usage, average training time and inference time of one epoch between BernPool and other baseline methods on the DD dataset in Table C. For quantitative analysis of efficiency, we compare the computation complexity between our methods and other pooling methods in Table D. Besides, we reclarify the stability of our BernPool from the theoretical perspective (in the supplementary material) and also provide the experimental evaluation.

These evaluation results demonstrate that the rationality of motivation and our BernPool is simple and efficient. Here we restate our contribution: 1) propose a probabilistic Bernoulli sampling method to not only learn effective sampling but also preserve high efficiency; 2) propose a Bernoulli-induced hybrid graph pooling way to retain both those sampled substructures and the remaining info; 3) verify the effectiveness and high-efficiency, and report the state-of-the-art performance.



_BernPool may be regarded as a plug-and-play component, and we have fully released the code, which could directly run the used ten datasets including tested ones by the rebuttal._

---

### Decision · Program_Chairs · 2023-09-21

**Decision:**

Reject

**Comment:**

Reviewers acknowledged the contributions of this paper regarding algorithm design and paper writing. However, they still raise concerns regarding the experiments, the expressiveness of the model, and the novelty (i.e., the idea of considering probabilistic pooling). Based on their comments, this paper seems to be borderline and it may need further revisions regarding the motivation and experiments to make the benefit of the proposed method more convincing.